# Comparative Study on the Periodontal Parameters Used in Diagnosing Periodontitis in Puerperae and Periodontitis’ Relationship with the Birth of Preterm Infants: A Case-Control Study

**DOI:** 10.3390/ijerph21020156

**Published:** 2024-01-30

**Authors:** Nayra Rodrigues de Vasconcelos Calixto, Fernanda Ferreira Lopes, Marcela Mayana Pereira Franco, Isaac Suzart Gomes-Filho, Bruno Braga Benatti, Cláudia Maria Coêlho Alves

**Affiliations:** 1Graduate Program in Dentistry, Federal University of Maranhão, São Luís 65080-805, Brazilfernanda.ferreira@ufma.br (F.F.L.); marcela.franco@ufma.br (M.M.P.F.); bruno.benatti@ufma.br (B.B.B.); 2Department of Dentistry II, Federal University of Maranhão, São Luís 65080-805, Brazil; 3Department of Health, State University of Feira de Santana, Feira de Santana 44036-900, Brazil; isgfilho@uefs.br

**Keywords:** pregnancy complications, premature infant, periodontal diseases, clinical diagnosis

## Abstract

To compare different criteria for the diagnosis of periodontitis and to evaluate the association of this condition with prematurity, this case-control study was conducted on 283 mothers of infants, divided into two groups based on gestational age (cases: <37 weeks, controls: ≥37 weeks), with 71 cases and 212 controls. The periodontal evaluation included probing depth (PD), clinical attachment level (CAL), plaque index, and bleeding on probing (BOP). Participants were classified regarding periodontitis per 14 criteria based on different periodontal parameters. The criterion selected as the gold standard was the presence of at least four teeth with one or more sites with a PD ≥ 4 mm, CAL ≥ 3 mm, and BOP at the same site. The prevalence of periodontal disease ranged from 8.1% to 55.1%. Moreover, compared to the gold standard, the sensitivities of the other criteria were 100%, while specificity ranged from 50.4% to 96.4%. Periodontitis, defined by six of the selected criteria, was associated with prematurity after multivariate adjustment, with OR ranging from 1.85 to 2.69 and 95% CI from 1.01 to 5.56; one of them was the gold standard mentioned above. Measurements using the clinical parameters of PD, CAL, and bleeding at the same site (criteria 5, 6, 7, 8), CPI (criterion 10), and at least four teeth with a PD ≥ 4 mm and CAL ≥ 3 mm (criterion 11) to define periodontitis showed a statistically significant association (*p* < 0.05). Given this study’s limitations, we can conclude that the diagnostic criteria for a periodontitis definition using a PD ≥ 4 mm and CAL ≥ 3 mm in two or more teeth, with BOP at the same site, seem stronger when detecting an association between periodontitis and prematurity.

## 1. Introduction

Periodontitis is an oral dysbiotic disease mediated by the host’s inflammatory response and characterized by the progressive destruction of tooth-supporting tissue, resulting in the loss of periodontal attachment and, consequently, marginal loss of alveolar bone [1]. Some research and a systematic review of the possible relationship between periodontitis and prematurity have shown controversial results. Some studies find a positive association, and others do not [2,3,4,5,6]; these results may be due to different sample sizes and diagnostic criteria, for example. Diagnostic criteria for periodontitis vary widely, which can lead to unsatisfactory internal validity, interfering with the research results regarding the association between periodontitis and prematurity and compromising comparability and reliability because of the complex clinical characteristics of periodontitis [7].

According to the World Health Organization (WHO), premature birth occurs before the 37th week of pregnancy. Behavioral factors, psychosocial risk, environmental exposure, medical, biological and genetic factors, and socioeconomic conditions can be associated with preterm birth [8]. Prematurity is a public health issue and a primary cause of morbimortality among newborns, compromising their health and leading to sequelae in adulthood [9].

A preterm birth can be related to intrauterine infections and to elevated levels of local or systemic inflammatory mediators through a reservoir of bacteria ascending from the vagina or through the hematogenous route to reach the placenta [8]. There still is a possible relationship between periodontitis, premature birth, and low birth weight due to the spread of periodontopathogenic bacteria and their products through the hematogenous route, triggering premature labor and low birth weight [2].

No uniform standard diagnostic criteria have been used in studies to define periodontitis [10]. The existing diagnostic criteria were not used with pregnant women but only with patients in general. Therefore, it is challenging to select a criterion for classifying periodontal disease and compare it with scientific findings in this area, specifically regarding the prematurity outcome. Therefore, the present study aimed to compare the diagnostic criteria for periodontitis to a gold standard criterion, identify their sensitivity, specificity, and positive and negative predictive values, and determine these criteria’s influence on estimating the association between periodontitis and prematurity.

## 2. Materials and Methods

### 2.1. Study Design and Sample Selection

This case-control study was conducted at the University Hospital of the Federal University of Maranhão, Maternal and Child Unit (HUUFMA-UMI), São Luís (MA), Brazil, from October 2011 to December 2012. The research was approved by the Research Ethics Committee of HUUFMA (Protocol No. 002673/2011-60). The sample consisted of women up to 48 h postpartum, and all subjects provided written informed consent to participate in the study. Women diagnosed with HIV or sickle cell anemia and women who required antibiotic prophylaxis for dental treatment or underwent periodontal treatment during pregnancy were excluded.

The case group consisted of women with infants born alive after less than 37 full gestational weeks (preterm), and the control group consisted of women with born-alive term infants (≥37 weeks of gestation). Mothers with infants born after less than 37 full gestational weeks who were in the institution after delivery were invited to participate in the study. The control group was obtained from the same period and source, comprising mothers with born term infants, selected randomly. (Figure 1) This is a case-control study because the comparisons of the different diagnostic criteria for periodontitis with a gold standard criterion are all related to the presence or absence of prematurity.

A posterior statistical power calculation was performed assuming a case-control ratio of 1:3, an alpha error of 0.05, a two-tailed test, odds ratio, and proportions obtained with criterion 8 [11]. Criterion 8 was defined as having at least four or more teeth with one or more sites with a PD ≥ 4 mm, CAL ≥ 3 mm at the same site, and the presence of BOP [11]. The 71 cases and 212 controls sample size were sufficient to ensure 91.12% power to estimate significant differences [11]. Selecting the criterion of 1 case per 3 controls was justified because the studied event (prematurity) is uncommon; therefore, given the limited possibility of finding cases, studies indicate that the inclusion of a more significant number of controls provides a more accurate estimate of the frequency of exposure in the control group and may increase the study’s power [12].

The covariates adopted in this study were age, household income, educational level, social class, previous obstetric history, previous preterm delivery, previous low-weight delivery, arterial hypertension, eclampsia, hyperemesis gravidarum, and urinary infection. Covariates obstetric history, previous preterm delivery, previous low-weight delivery, arterial hypertension, eclampsia, hyperemesis gravidarum, and urinary infection were classified by presence or absence. The women’s ages were classified as 18–20 years, 21–35 years, or 36–43 years. Household income was classified as up to one minimum wage or more than one minimum wage. The educational level was classified as 0–8 years of schooling or more than 8 years of schooling. According to the Brazilian economic criterion, the covariate social class was classified as B, C, D, or E.

### 2.2. Periodontal Evaluation

A single investigator, a previously trained periodontics specialist, who was blinded to the birth conditions, performed the periodontal examinations throughout the study. The calibration process was performed per WHO recommendations; 10 patients were examined at two different times, one week apart, at the Dental Clinic of the Federal University of Maranhão [13]. Participants also answered a structured questionnaire with information about maternal oral health habits, access to dental services, maternal socioeconomic characteristics, and harmful habits. The socioeconomic level was assessed under the Brazilian economic classification criterion proposed by the Brazilian Association for Market Research [14]. Data such as gestational age, current and previous gestational history, and factors related to their general health status were collected from the medical records. The periodontal examination was performed in the hospital bed within 48 h postpartum, with the subject in a sitting position under artificial lighting, using forceps, a mouth mirror, and a millimeter periodontal probe (North Carolina PCPUNC 15, Hu-Friedy©, Chicago, IL, USA). The following parameters were evaluated: (1) probing depth (PD), (2) clinical attachment level (CAL), (3) bleeding on probing (BOP) at six sites per tooth (the mesiobuccal and mesiolingual, distobuccal and distolingual, and mid-buccal and mid-lingual regions), and (4) plaque index (pi) at 4 sites per tooth (buccal, mesial, distal, and lingual) [15]. The PD was measured as the distance from the soft tissue margin to the tip of the periodontal probe [15]. The CAL was measured as the distance from the cemento-enamel junction to the tip of the periodontal probe [15]. BOP was determined by the presence or absence of blood after periodontal probing [15].

### 2.3. Periodontitis Diagnostic Criteria

Fourteen periodontitis diagnostic criteria were selected based on previous studies that associated periodontitis criteria with prematurity. The diagnosis was conducted according to the presence or absence of periodontitis. Criterion 1: at least one site with CAL ≥ 3 mm and PD ≥ 4 mm [16]. Criterion 2: at least two sites with CAL ≥ 3 mm and PD ≥ 4 mm [16]. Criterion 3: at least three sites with CAL ≥ 3 mm and PD ≥ 4 mm [17]. Criterion 4: at least four sites with CAL ≥ 3 mm and PD ≥ 4 mm [16]. Criterion 5: at least one tooth with one or more sites with PD ≥ 4 mm, with CAL ≥ 3 mm at the same site, and BOP [17]. Criterion 6: at least two teeth with one or more sites with PD ≥ 4 mm, with CAL ≥ 3 mm at the same site, and the presence of BOP [16]. Criterion 7: at least three teeth with one or more sites with PD ≥ 4 mm, with CAL ≥ 3 mm at the same site, and the presence of BOP [16]. Criterion 8: at least four or more teeth with one or more sites with PD ≥ 4 mm, with CAL ≥ 3 mm at the same site and presence of BOP [11]. Criterion 9: at least three or more teeth with a site with CAL ≥ 3 mm [17]. Criterion 10: use the Community Periodontal Index (CPI) and evaluate index teeth, considering the tooth with the deepest PD per sextant. Periodontitis: at least one site with PD ≥ 4 mm [18]. Criterion 11: at least four teeth with one or more sites with PD ≥ 4 mm and with CAL ≥ 3 mm at the same site [19]. Criterion 12: CAL ≥ 4 mm [20]. Criterion 13: at least one or more sites with PD ≥ 4 mm and 50% of teeth with BOP [21]. Criterion 14: one tooth with a PD site and CAL ≥ 4 mm [22]. The gold standard adopted was Criterion 8 [11].

### 2.4. Statistical Analysis

Data were analyzed using the SPSS software (version 28.0; IBM, Chicago, IL, USA) and GraphPad Prism software (version 9.1; GraphPad Software, San Diego, CA, USA). The normality assumption was tested using the Shapiro–Wilk test. Most of the continuous variables violated the assumptions of normality, so the Mann–Whitney test was employed to compare the periodontal parameters between groups. The chi-square or Fisher’s exact test was adopted to analyze the distribution of frequencies between the case and control groups. The crude odds ratio (OR) and respective 95% confidence interval (95% CI) were calculated to estimate the strength of the association between the exposure variables and prematurity outcome. Multivariate logistic regression analysis was used to estimate the adjusted OR between periodontitis and prematurity. Covariates with a *p*-value < 0.10 in univariate analysis (maternal education level, social class, and arterial hypertension) were included in this model as confounders. We used a multiple linear regression model to evaluate a multiple linear regression model in order to assess whether the confounding factors of prematurity were associated with CAL. The models’ assumptions were verified based on the absence of collinearity between independent variables. None of the variables suffered from multicollinearity.

The prevalence of periodontitis was calculated per criterion chosen for the study. The report by Gomes Filho et al. (2007) was adopted as the gold standard [11]. Thus, we calculated sensitivity, specificity, positive and negative predictive values, and respective 95% CIs. We selected the gold standard based on the specificity of the criterion and the consistent association between periodontitis and prematurity compared to the association obtained between other periodontitis diagnostic criteria and the same outcome. The level of significance was set at 0.05 for all statistical tests.

## 3. Results

The study sample comprised 283 participants, 71 cases, and 212 controls. Mothers were aged between 18 and 43 years and the most prevalent in the studied groups were aged 21–35 years. Only the arterial hypertension covariate showed a higher frequency among cases compared to controls with a statistically significant difference (*p* < 0.001) (Table 1). There was no statistically significant difference (*p* = 0.89) between the case (23.5 ± 4.5) and control (23.7 ± 4.6) groups concerning the pregestational BMI summary measurement (mean ± standard deviation).

Figure 2 shows the means and standard deviations of the periodontal variables. The case group had the worst periodontal conditions compared to the control group: a higher percentage of teeth with PD ≥ 4 mm, teeth with CAL ≥ 3 mm, BOP index, and plaque index. The difference was statistically significant for the percentage of teeth with PD ≥ 4 mm (*p* < 0.006). Additionally, the multivariate regression analysis for CAL showed that arterial hypertension was related to higher levels of CAL in the evaluated sample (estimative = 3.49, SE = 1.02, *p* <0.001, Table 2).

In the crude measurement analysis, most exposure diagnostic criteria showed a significant positive association, except for those that used criteria 3, 9, and 13 to define the exposure (*p* ≥ 0.05). After adjusting for the confounders, educational level, social class, and arterial hypertension, 6 of the 14 measurements showed a statistically significant positive association between periodontitis and prematurity. However, these results should be interpreted with caution because some lower limits of the confidence interval are at the borderline (criteria 5, 10, 11). Measurements using the clinical parameters of PD, CAL, and bleeding on probing at the same site (criteria 5, 6, 7, 8), CPI (criterion 10), and at least four teeth with a PD ≥ 4 mm and CAL ≥ 3 mm (criterion 11) to define periodontitis showed a statistically significant association (*p* < 0.05) (Table 3).

The frequency of periodontitis diagnosis ranged from 8.1% (criterion 13) to 55.1% (criterion 9) per the criterion used (Table 4). A gradual decline in the percentage of diagnoses was observed from criteria 1 to 4 and 5 to 8 due to the increased number of affected teeth. The groups that used the presence of bleeding as an evaluation parameter (criteria 5 to 8 and 13) showed a lower frequency trend. Criteria 9 and 12, which consider at least one tooth with only a CAL ≥ 3 or 4 mm, respectively, and the criterion that uses only a few index teeth showed higher frequencies (Table 4).

Compared to the gold standard (criterion 8), all criteria, except 13, showed 100% sensitivity, indicating a high capacity to identify positive individuals. Criteria 4, 6, 7, 11, and 13 showed specificities above 90%, indicating a high capacity to identify individuals without the disease. The criteria with the highest positive predictive values were 7 (75.6%, 95% CI = 59.3–87.0) and 11 (72%, 95% CI = 56.1–84.2), revealing the percentage of participants with a positive diagnosis and periodontitis. The negative predictive value was 100% (95% CI = 96.3–100) for all criteria, except criterion 13 (93.4%, 95% CI = 89.5–96.0), revealing the proportion of women who tested negative and who did not have periodontitis (Table 4).

## 4. Discussion

This case-control study analyzed periodontal disease as an exposure for younger gestational age in 283 pregnant women and analyzed periodontal disease according to 14 different diagnostic criteria. According to 10 of the 13 criteria, periodontitis was associated with prematurity in the crude analysis, while in the adjusted analysis, according to 6 criteria, periodontitis was associated with the outcome. Therefore, the different criteria for diagnosing periodontitis may influence the prevalence of periodontitis and its association with prematurity.

Adjustments were performed for classic confounders such as hypertension, a significant risk factor for prematurity [23]. Hypertension was the most frequent finding in the cases, while the body mass index was equivalent between the two groups. Adjustments were also made for socioeconomic factors such as education and social class. A lack of adjustments or inadequate adjustments are essential methodological flaws [24].

In the sample cohort, mothers with periodontitis were twice as likely to give birth to a premature baby than those without periodontitis (*p* < 0.05) according to criteria 6, 7, 8, and 11. Prematurity was also assessed based on the most recent AAP diagnosis criteria, and a similar result was found [25,26]. A recent systematic review with meta-analysis found an association between periodontitis and other adverse neonatal outcomes such as pre-eclampsia, gestational diabetes, and low birth weight [10]. The explanation for the relationship is that periodontitis can activate cascades of inflammatory immune mediators, such as prostaglandin E2 (PGE2), IL-6, IL-1, and TNF-alpha, and thus be related to adverse perinatal outcomes. Furthermore, periodontitis can act as a source of bacteria. Then, inflammatory mediators are transferred through the oral cavity to the fetoplacental unit through the blood circulation, resulting in adverse effects.

The clinical parameters of criteria 5 to 8 shared a PD parameter ≥ 4 mm and the presence of BOP, which signals a current inflammatory load [27]. The presence of BOP combined with CAL and PD can be justified by the greater rigor obtained. This clinical parameter reinforces the specific characteristic of tissue inflammation in periodontitis [11]. The presence of ulceration in the epithelial lining of the soft tissue wall of the periodontal pocket could be adequate for reflecting systemic markers that represent tissue invasion and the systemic spread of periodontopathogenic bacteria and their byproducts [28]. BOP has already been used in other studies [11,29] as one of the diagnostic parameters for periodontitis, confirming the association between periodontitis and prematurity/low birth weight. However, other authors [30] argue that the presence of BOP may not be a good predictor of the severity or progression of periodontitis.

The CAL ≥ 3 mm parameter was common to several criteria (5 to 8 and 11). The CAL is characterized by reflecting the cumulative individual history of periodontitis. This measurement gauges the risk of outcomes arising from prolonged exposure to risk factors [28]. More frequent use of the CAL measurement for the diagnosis of periodontitis has been observed in previous studies, as it is the most accurate among existing clinical descriptors [17]. It has been considered the ‘gold standard’ for determining the history and progression of periodontitis. It is widely used in clinical studies, epidemiological surveys [31,32], and research on the association with systemic conditions such as prematurity [11,29].

The presence of a PD ≥ 4 mm and CAL ≥ 3 mm in two or more teeth, and preferably with BOP at the same site, are criteria associated with prematurity (criteria 6 to 8). Combining continuous clinical measurements of the disease categories is the best way to measure periodontitis when investigating its impact on pregnancy [29,33]. In the absence of knowledge about which signs of periodontitis would be adequate for measuring systemic damage, one would employ the signs that reveal the accumulated history of the disease (CAL) or those that show the current inflammatory load (PD, BOP) [28,30] in the search for a more rigid and specific criterion that can reduce false positives and distortions [11].

Another finding was the detection of two criteria with high sensitivity and specificity when using the criteria of Gomes-Filho et al. (2007) as a gold standard [11] for comparison. We selected the gold standard criterion [11] because it showed good specificity for association studies in a population group of pregnant women, in addition to including the clinical parameter of bleeding on probing, which is very common during pregnancy and gum inflammation.

The criterion that used at least three teeth, with one or more sites with a PD ≥ 4 mm, CAL ≥ 3 mm at the same site, and the presence of BOP (criterion 7) showed 100% sensitivity (95% CI = 86.3–100) and 96% specificity (95% CI = 92.6–97.9). Criterion 11, which used at least four teeth with one or more sites with PD ≥ 4 mm and CAL ≥ 3 mm at the same BOP site (criterion 7), showed 100% sensitivity (95% CI = 86.3–100) and 95.2% specificity (95% CI = 91.6–97.4).

It is essential to investigate specificity as it is a measure of diagnostic accuracy defined as the proportion of individuals or sites with genuinely absent disease who tested negative. The more specific the test, the less likely the patient is to be positive in the absence of disease, meaning that there are few or no false-positive values [22]. Similarly, sensitivity is the proportion of individuals or sites with a genuinely present disease. A susceptibility test is unlikely to fail to detect a disease. However, it is not uncommon to detect false-positive results. It is also known that sensitivity is lost when choosing a high-specificity test since some cases in borderline situations can be classified as non-sick [11].

The classification of the American Academy of Periodontology and the European Federation of Periodontology [1] was proposed as a final diagnostic criterion. However, according to this criterion, all women were diagnosed with periodontitis, which is due to the socioeconomic level of the sample and the number of missing teeth. Furthermore, no radiographic examination was performed because we could not expose the postpartum woman to radiation during breastfeeding and because of the difficulty for hospitalized women in traveling to a radiology center.

Besides not adopting the current classification, another limitation of this study is the difficulty in discussing the clinical diagnostic parameters employed since we could not identify studies comparing the different diagnostic criteria for periodontal disease, except for the study by Gomes Filho. Among the strengths, it is essential to highlight that this is one of the few studies on the adjustment of the analysis, the investigation of the specificity and sensitivity of the different criteria for diagnosing periodontitis, and the association of prematurity with the adoption of various definitions for periodontal disease, expanding the discussion of periodontal disease as an essential exposure for prematurity. Another aspect that should be highlighted is using three controls for one case. A study recommends the adoption of two or three controls in cases like our study when the gain in precision with a more significant number would not be relevant [34].

## 5. Conclusions

In conclusion, different diagnostic parameters can influence the prevalence of periodontitis and its association with systemic conditions such as preterm birth. Given this study’s limitations, diagnostic criteria using a PD ≥ 4 mm and CAL ≥ 3 mm in two or more teeth, preferably with BOP at the same site, seem more appropriate for detecting an association between periodontitis and preterm birth. These criteria also show greater sensitivity and specificity when compared to the gold standard criteria.

## Figures and Tables

**Figure 1 ijerph-21-00156-f001:**
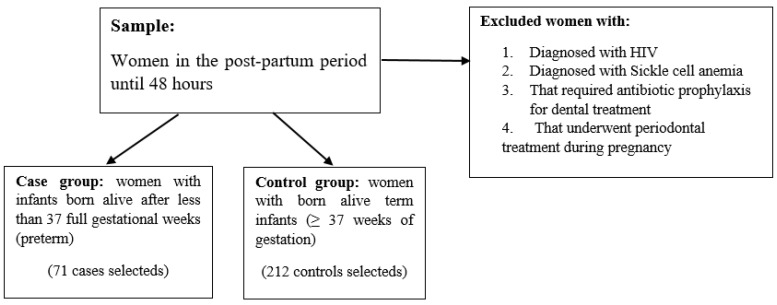
Sample selection flowchart.

**Figure 2 ijerph-21-00156-f002:**
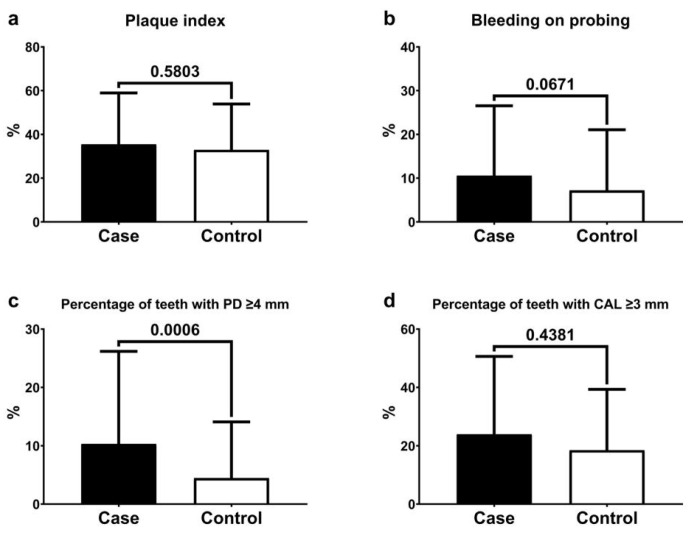
Means and standard deviations of plaque index (**a**), bleeding on probing (**b**), percentage of teeth with a probing depth (PD) ≥ 4 mm (**c**), percentage of teeth with a clinical attachment level (CAL) ≥ 3 mm (**d**), and comparative analyses between the case and control groups.

**Table 1 ijerph-21-00156-t001:** Distribution of the investigated covariates between case and control groups.

Covariates	Case		Control		*p*-Value
	*n*	%	*n*	%	
Sociodemographic factors					
Age group					0.406
18 to 20 years	12	16.9	51	24.1	
21 to 35 years	52	73.2	138	65.1	
36 to 43 years	7	9.9	23	10.8	
Household income					0.462
Up to one minimum wage	32	48.5	87	43.3	
More than one minimum wage	34	51.5	114	56.7	
Educational level					0.054
0–8 years of schooling	25	35.2	50	23.6	
More than 8 years of schooling	46	64.8	162	76.4	
Social class ^a^					0.086
B	6	8.5	30	14.2	
C	40	56.3	132	62.9	
D and E	25	35.2	48	22.9	
Previous obstetric history					
Parity					0.887
Primiparous	32	45.1	93	44.1	
Multiparous	39	54.9	118	55.9	
Previous preterm delivery (<37 weeks)					0.225
Yes	8	24.2	17	15.2	
No	25	75.8	95	84.8	
Previous low weight delivery (<2500 g)					0.190
Yes	8	24.2	16	14.5	
No	25	75.8	94	85.5	
Current obstetric history					
Arterial hypertension					<0.001 *
Yes	33	46.5	40	19.0	
No	38	53.5	171	81.0	
Eclampsia					1.000
Yes	2	2.8	5	2.4	
No	69	97.2	207	97.6	
Hyperemesis gravidarum					0.120
Yes	6	9.0	8	4.0	
No	61	91.0	194	96.0	
Urinary infection					1.000
Yes	23	32.9	69	32.5	
No	47	67.1	143	67.5	

^a^ According to the Brazilian Economic Criterion. * *p* < 0.05.

**Table 2 ijerph-21-00156-t002:** Multivariate regression model for CAL data.

Predictor	Estimated	Standard Error	*p*-Value
Educational level (Reference: ≤8 years)			
>8 years of schooling	−1.42	1.06	0.181
Arterial hypertension (reference: No)			
Yes	3.49	1.02	<0.001
Social (reference: B class)			
C class	3.88	1.97	0.051
D-E class	2.7	2.37	0.257

**Table 3 ijerph-21-00156-t003:** Associations [crude and adjusted odds ratio (OR) and 95% confidence interval (95% CI)] between periodontitis and prematurity according to the adopted criteria.

Diagnostic Criteria	Periodontitis Frequency (%)		Crude OR (95% CI)	*p*-Value	Adjusted OR ^a^ (95% CI)	*p*-Value
	Case	Control				
Criterion 1	49.3	32.1	2.05 (1.19–3.55)	0.009 *	1.76 (0.98–3.14)	0.054
Criterion 2	40.8	24.5	2.12 (1.20–3.74)	0.008 *	1.65 (0.90–3.02)	0.100
Criterion 3	32.4	21.2	1.77 (0.97–3.22)	0.056	1.38 (0.73–2.61)	0.317
Criterion 4	40.0	15.1	2.52 (1.34–4.73)	0.003 *	1.95 (0.99–3.80)	0.050
Criterion 5	39.4	21.7	2.34 (1.31–4.18)	0.003 *	1.85 (1.01–3.43)	0.049 *
Criterion 6	35.2	13.7	3.42 (1.83–6.40)	<0.001 *	2.55 (1.30–4.99)	0.006 *
Criterion 7	28.2	9.9	3.56 (1.79–7.08)	<0.001 *	2.69 (1.30–5.56)	0.007 *
Criterion 8 ^b^	22.5	7.1	3.82 (1.77–8.21)	<0.001 *	2.60 (1.14–5.89)	0.021 *
Criterion 9	57.7	54.2	1.15 (0.66–1.98)	0.607	0.89 (0.50–1.60)	0.720
Criterion 10	45.1	27.4	2.17 (1.24–3.80)	0.005 *	1.89 (1.05–3.42)	0.032 *
Criterion 11	26.7	11.3	2.86 (1.45–5.62)	0.001 *	2.10 (1.01–4.33)	0.044 *
Criterion 12	49.3	34.4	1.85 (1.07–3.19)	0.025 *	1.46 (0.82–2.61)	0.195
Criterion 13	11.3	7.1	1.66 (0.67–4.11)	0.263	1.44 (0.55–3.74)	0.451
Criterion 14	47.9	30.7	2.07 (1.19–3.60)	0.008 *	1.77 (0.99–3.17)	0.051

^a^ Adjusted for the following covariates: educational level, social class, and arterial hypertension. ^b^ the gold standard adopted * *p* < 0.05.

**Table 4 ijerph-21-00156-t004:** Measurements of diagnostic accuracy and frequency of periodontitis obtained with the different diagnostic criteria compared to the gold standard (Criterion 8).

Diagnostic Criteria	*n*	%	Sensitivity (%) (95% CI)	Specificity (%) (95% CI)	PPV (%) (95% CI)	NPV (%) (95% CI)
Criterion 1	103	36.4	100 (86.3–100)	71.4 (65.3–76.8)	30.1 (21.6–40.0)	100 (97.4–100)
Criterion 2	81	28.6	100 (86.3–100)	80.1 (74.5–84.7)	38.3 (27.9–49.8)	100 (97.7–100)
Criterion 3	68	24.0	100 (86.3–100)	85.3 (80.2–89.3)	45.6 (33.6–58.0)	100 (97.8–100)
Criterion 4	54	19.1	100 (86.3–100)	90.9 (86.4–94.0)	57.4 (43.3–70.5)	100 (97.9–100)
Criterion 5	74	26.2	100 (86.3–100)	82.9 (77.6–87.2)	41.9 (30.7–53.9)	100 (97.7–100)
Criterion 6	54	19.1	100 (86.3–100)	90.9 (86.4–94.0)	57.4 (43.3–70.5)	100 (97.9–100)
Criterion 7	41	14.5	100 (86.3–100)	96.0 (92.6–97.9)	75.6 (59.3–87.0)	100 (98.0–100)
Criterion 9	156	55.1	100 (86.3–100)	50.4 (44.1–56.7)	19.9 (14.1–27.1)	100 (96.3–100)
Criterion 10	90	31.8	100 (86.3–100)	76.6 (70.8–81.6)	34.4 (24.9–45.3)	100 (97.6–100)
Criterion 11	43	15.2	100 (86.3–100)	95.2 (91.6–97.4)	72.0 (56.1–84.2)	100 (98.0–100)
Criterion 12	108	38.2	100 (86.3–100)	69.4 (63.2–74.9)	28.7 (20.6–38.3)	100 (97.3–100)
Criterion 13	23	8.1	45.2 (27.7–63.7)	96.4 (93.1–98.2)	60.8 (38.7–79.5)	93.4 (89.5–96.0)
Criterion 14	99	35.0	100 (86.3–100)	73.0 (67.0–78.3)	31.3 (22.6–41.5)	100 (97.4–100)
Criterion 8 ^b^	31	10.9				

95% CI = 95% confidence interval; PPV = positive predictive value; NPV = negative predictive value. ^b^ the gold standard utilized

## Data Availability

The authors confirm that data supporting this study’s findings are available within the article.

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
