# Peer review of "Comparative Study on the Periodontal Parameters Used in Diagnosing Periodontitis in Puerperae and Periodontitis’ Relationship with the Birth of Preterm Infants: A Case-Control Study"

_ijerph, 2024, doi:10.3390/ijerph21020156_

Round 1

Reviewer 1 Report

Comments and Suggestions for Authors

I thank authors for conducting well designed clinical case controlled study. I have the following concerns:

Introduction: (see yellow highlights in the pdf file)

1.    Definition must be redefined and references: Periodontitis is a chronic multifactorial inflammatory disease associated with dysbiotic plaque biofilms and characterized by progressive destruction of the tooth-supporting apparatus. 

2.    The statement is not clear, re-phrase. 

3.    The new classification (2017) must be stated (Mentioned in the last paragraph in discussion). Also what make authors believe `Filho et.al 2007 is the gold standard?

Methods:

4.    Flow chart should be included.

5.    Can you be more specific; 0ne month or more postdelivery?

6.    The sample size was stated and justified. I don’t see it necessary to be included. It may be included in the discussion section. 

7.    14 criteria were chosen; and the gold standard was not specified. It would be better for readers to include it as table 

Results:

8.    The age stated (21-35), yet in table 1, it was 12-43, can you clarify?

9.    It is possible to include regression analysis for CAL data after controlling cofounding factors.

Author Response

Introduction

Comment:

  1. Definition must be redefined and references: Periodontitis is a chronic multifactorial inflammatory disease associated with dysbiotic plaque biofilms and characterized by progressive destruction of the tooth-supporting apparatus.

Response:

Line 34-37. We appreciate the suggestion, and we have made changes in the text.

Comment:

  1. The statement is not clear, re-phrase. 

Response:

Line 57-58. We appreciate the suggestion and made changes in the text.

Comment:

  1. The new classification (2017) must be stated (Mentioned in the last paragraph in discussion). Also what make authors believe `Filho et.al 2007 is the gold standard?

Response:

We appreciate the reviewer's comment and as requested, we have included additional information about the 2018 Classification of Periodontal Diseases in the last paragraph of the discussion (line 305 to 311) (Tonetti et al., 2018). Furthermore, as this classification is very sensitive for our study sample since women have a low socioeconomic status and all have tooth loss, it was not possible to use it in this study as we would not have a comparison group, i.e. that is, women without periodontitis. When classifying according to the criteria mentioned above, all participants were diagnosed with Periodontitis. Therefore, the criterion of Gomes-Filho et al, 2007 was chosen as the gold standard, since its good specificity has already been proven for a study with a population group of pregnant women, in addition to including the clinical parameter of bleeding on probing, which is very common during pregnancy, gingival inflammation. Therefore, the following information was incorporated into the Discussion section:

As a final diagnostic criterion, it was proposed to use the classification of the American Academy of Periodontology and European Federation of Periodontology [1], however, according to this criterion, all women were diagnosed with periodontitis. This is due to the socioeconomic level of the sample and the number of missing teeth. Furthermore, no radiographic examination was performed because we could not expose the postpartum woman to radiation during breastfeeding and because of the difficulty for hospitalized women in traveling to a radiology center.

In addition to not using the current classification, another limitation that can be pointed out is the difficulty in discussing the clinical diagnostic parameters used, since no studies are found that compare the different diagnostic criteria for periodontal disease, except for the study by Gomes Filho. Among the strengths, it is essential to highlight that this is one of the few studies that made an adjustment of the analysis, an investigation of the specificity and sensitivity of the different criteria for diagnosing periodontitis, and the association of prematurity with the adoption of various definitions for periodontal disease, expanding the discussion of periodontal disease as an important exposure for prematurity.

 Methods

Comment:

  1. Flowchart should be included.

Response:

Line 107. We appreciate the suggestion, and we included the flowchart in the manuscript.

Comment:

  1. Can you be more specific; 0ne month or more postdelivery?

Response:

Line 72. We apologize for the mistake. The correct information was incorporated into the text, as follows: The sample consisted of women up to 48 hours of its postpartum period, and all.

Comment:

  1. The sample size was stated and justified. I don’t see it necessary to be included. It may be included in the discussion section.

Response:

Line 321-322. We appreciate the suggestion. We removed the paragraph and we added to the discussio on line 321-322.

Comment:

  1. 14 criteria were chosen; and the gold standard was not specified. It would be better for readers to include it as table.

Response:

Line 148-149. We appreciate the suggestion. It was included in the item 2.3 on the line 147-148 and in Tables 2 and 3.

Results

Comment:

  1. The age stated (21-35), yet in table 1, it was 12-43, can you clarify?

Response:

We appreciate the suggestion and we have made changes as suggested. A description of the age ranges was included in the results and the age range was changed to 18 to 43 years, since patients between 12 and 17 were excluded from the sample.

Comment:

  1. It is possible to include regression analysis for CAL data after controlling cofounding factors.

Response:

Line 185-187. We appreciate the suggestion. We have included a multiple linear regression model to evaluate and assess whether confounding factors of prematurity were associated with CAL. We have added the analysis according to reviewer’s recommendation (new table 2).

Reviewer 2 Report

Comments and Suggestions for Authors

“Comparative study of periodontal parameters used in the diagnosis of periodontitis in puerperal and its relationship with the 3 births of preterm infants” was submitted to IJERPH.

This study aimed to compare different criteria for the diagnosis of periodontitis and to evaluate the association of this condition with prematurity.

The authors concluded that the diagnostic criteria for periodontitis definition using PD ≥ 4 mm and CAL ≥ 3 mm in two or more teeth, with BOP at the same site, seem more consistent when detecting an association between periodontitis and prematurity.

The manuscript deals with an interesting issue; however, there are several concerns related to the study.

Title: Please include the type of study.

Abstract

Line 15.  Please indicate the number of cases and the number of controls.

Line 20. Please be specific with the periodontal diagnosis.

Lines 22-23. Does periodontitis defined by the six mentioned criteria correspond to the gold standard defined above? Please clarify and comment on this.

Line 23. Please present the most relevant significant odds ratios from the multivariate analysis.

Line 25. What do you mean by 'seem more consistent'? Please provide more specific details.

Keywords: Make sure they correspond to MeSH terms.

Introduction

In the definition of periodontitis, include the concept of dysbiosis.

Line 34. Briefly describe the factors that lead to these controversial results.

Lines 51-52. This concept is based on an outdated reference related to a diagnostic classification of periodontitis.

Methods

Please describe the patient recruitment dates.

The study design is not clear. Merely presenting case patients and control patients does not necessarily indicate a case-control study. Please provide details on the elements that established this design.

Line 74. Please provide details about what criterion 8 refers to.

Line 83. Please provide details about the operator calibration process, the statistical tests used to determine calibration, and the results.

Lines 89 and 91. Please review references 13 and 14 as they do not appear to relate to what is described in the text.

The same issue is present with references 15, 16, and 17. This creates significant problems in verifying the accuracy and reliability of the diagnostic criteria. Nevertheless, it can be observed that outdated diagnostic references are being used.

The criteria presented and supported by references 7, 15, 18, 22, and 23 are outdated. This will likely cause significant confusion among readers. It is important to explain the reasons for not considering the criteria established by the current classification of periodontitis.

Line 123. It appears that the data did not follow a normal distribution. Please provide the test used to determine this and its p-value.

Line 127. Please indicate how the model assumptions were verified.

Line 132. This criterion was published more than 15 years ago. Furthermore, this criterion corresponds to a publication by the same authors of this manuscript. This aspect raises concerns about objectivity. It is essential to clearly explain the reasons for not considering the current classification.

Results

Line 142. Table 1 should be presented before Figure 1.

Lines 145-148. These results are quite striking and should be adequately explained and compared in the discussion.

The size of Figure 1 should be increased for better visibility.

Line 158. The authors are combining ORs and confidence intervals from different criteria. This is not correct, and moreover, the upper limit of the confidence interval is incorrect. Please review this. Furthermore, the lower limit of this confidence interval is at the borderline; therefore, these results should be interpreted with caution. A similar situation occurs with criteria 5, 10, and 11(Table 2).

This should be detailed in the abstract and explained and discussed further in the main text.

Discussion

Line 189, 211 and 218. Please consider the observations made about these results.

Please consider the other aspects described above that need to be discussed. Additionally, more study limitations should be presented.

Line 244. This is not entirely accurate. Please review some studies on this topic published after the dissemination of the current classification.

Conclusions

Line 252. Please provide more specific details.

Some references are very outdated; therefore, it is recommended to replace them with current concepts.

Comments on the Quality of English Language

moderate

Author Response

1.Comment:

Title: Please include the type of study.

Response:

Line 4. We appreciate the suggestion and we have made changes in the text.

Abstract

2.Comment:

Line 15.  Please indicate the number of cases and the number of controls.

Response:

Line 16. We appreciate the suggestion and we have made changes as suggested, there are 71 cases and 212 controls.

3.Comment:

Line 20. Please be specific with the periodontal diagnosis.

Response:

Line 20. We appreciate the suggestion, and we rewrote the sentence.

4.Comment:

Lines 22-23. Does periodontitis defined by the six mentioned criteria correspond to the gold standard defined above? Please clarify and comment on this.

Response:

Six of the 14 criteria were associated with the outcome of prematurity. Among them is the criterion classified as the gold standard. This was included in the text.

5.Comment:

LINE 23. please present the most relevant significant odds ratios from the multivariate analysis.

Response:

Line 24-28. We appreciate the suggestion and we have made changes in the text.

6.Comment:

Line 25. What do you mean by 'seem more consistent'? Please provide more specific details.

Response:

Line 29. Thank you for the suggestion and we have made changes to improve understanding of the sentence.

  1. Comment:

Keywords: Make sure they correspond to MeSH terms.

Response:

Line 31. We have been checked, and we changed it according with the terms of the MeSH.

Introduction

  1. Comment:

In the definition of periodontitis, include the concept of dysbiosis.

Response:

Line 34-37. We appreciate the suggestion and we have made changes in the text.

  1. Comment:

Line 34. Briefly describe the factors that lead to these controversial results.

Response:

Line 39-40. We appreciate the suggestion and we have made changes in the text.

  1. Comment:

Lines 51-52. This concept is based on an outdated reference related to a diagnostic classification of periodontitis.

Response:

Line 57-58. We appreciate the suggestion and we have made changes in the text.

 Methods

11.Comment:

Please describe the patient recruitment dates.

Response:

Line 70. We appreciate the suggestion and we have included it in the text that the data for this study was found between October 2011 and December 2012.

  1. Comment:

The study design is not clear. Merely presenting case patients and control patients does not necessarily indicate a case-control study. Please provide details on the elements that established this design.

Response:

Line 81-83. We appreciate the suggestion and we have made changes in the text.

  1. Comment:

Line 74. Please provide details about what criterion 8 refers to.

Response:

Line 86-87. We appreciate the suggestion and we have made changes in the text.

  1. Comment:

Line 83. Please provide details about the operator calibration process, the statistical tests used to determine calibration, and the results.

Response:

Line 110-112. We appreciate the comment, the calibration process was carried out in accordance with World Health Organization (WHO) recommendations, with 10 patients being examined at two different times, one week apart, at the Dental Clinic of the Federal University of Maranhão.

  1. Comment:

Lines 89 and 91. Please review references 13 and 14 as they do not appear to relate to what is described in the text.

Response:

Line 117-119. We appreciate the suggestion and we have included the correct reference.

  1. Comment:

The same issue is present with references 15, 16, and 17. This creates significant problems in verifying the accuracy and reliability of the diagnostic criteria. Nevertheless, it can be observed that outdated diagnostic references are being used.

Response:

Line 121-128. We appreciate the suggestion and we have made changes as suggested, we included a new reference.

  1. Comment:

The criteria presented and supported by references 7, 15, 18, 22, and 23 are outdated. This will likely cause significant confusion among readers. It is important to explain the reasons for not considering the criteria established by the current classification of periodontitis.

Response:

We appreciate the reviewer's comment. We have included additional information about the 2018 Classification of Periodontal Diseases in the last paragraph of the discussion (line 305 to 311) (Tonetti et al., 2018). It is important to highlight that the new classification is very sensitive for our study sample since women have a low socioeconomic status and all have tooth loss, it was not possible to use it in this study as we would not have a comparison group, i.e. that is, women without periodontitis. Therefore, the following information was incorporated into the Discussion section at line 302-317:

As a final diagnostic criterion, it was proposed to use the classification of the American Academy of Periodontology and European Federation of Periodontology [1], however, according to this criterion, all women were diagnosed with periodontitis. This is due to the socioeconomic level of the sample and the number of missing teeth. Furthermore, no radiographic examination was performed because we could not expose the postpartum woman to radiation during breastfeeding and because of the difficulty for hospitalized women in traveling to a radiology center.

In addition to not using the current classification, another limitation that can be pointed out is the difficulty in discussing the clinical diagnostic parameters used, since no studies are found that compare the different diagnostic criteria for periodontal disease, except for the study by Gomes Filho. Among the strengths, it is essential to highlight that this is one of the few studies that made an adjustment of the analysis, an investigation of the specificity and sensitivity of the different criteria for diagnosing periodontitis, and the association of prematurity with the adoption of various definitions for periodontal disease, expanding the discussion of periodontal disease as an important exposure for prematurity.

  1. Comment:

Line 123. It appears that the data did not follow a normal distribution. Please provide the test used to determine this and its p-value.

Response:

Line 153-155. We appreciate the reviewer’s comment. We have rewritten the description of the statistical analysis to clarify this point. “The normality assumption was tested using the Shapiro-Wilk test. Most of the continuous variables violated the assumptions of normality, so the Mann-Whitney test was used to compare the periodontal parameters between groups.

  1. Comment:

Line 127. Please indicate how the model assumptions were verified.

Response:

Line 164-166. We appreciate the suggestion and we have made changes in the text.

  1. Comment:

Line 132. This criterion was published more than 15 years ago. Furthermore, this criterion corresponds to a publication by the same authors of this manuscript. This aspect raises concerns about objectivity. It is essential to clearly explain the reasons for not considering the current classification.

Response:

We appreciate the reviewer's comment and as requested, the criterion of Gomes-Filho et al, 2007 was chosen as the gold standard, since its good specificity has already been proven for a study with a population group of pregnant women, in addition to including the clinical parameter of bleeding on probing, which is very common during pregnancy, gingival inflammation. Furthermore, a most recent publication of this criterion was updated by Gomes-Filho et al. in 2018*. We kept the original reference to enhance the diagnostic criteria.

* GOMES-FILHO, I. S. et al. Clinical diagnostic criteria for periodontal disease: an update. Journal of Dental Health, Oral Disorders & Therapy, vol. 9, 14 Sep. 2018.

Therefore, the following information was incorporated into the Discussion section:

“the choice of the gold standard criterion (Gomes Filho et al., 2007) was due to the fact that this criterion has proven to have good specificity for association studies in a population group of pregnant women, in addition to including the clinical parameter of bleeding to probing, given that gingival inflammation is very common during pregnancy.”

Results

  1. Comment:

Line 142. Table 1 should be presented before Figure 1.

Response:

Line 188. We appreciate the suggestion and we have made changes in the text.

  1. Comment:

Lines 145-148. These results are quite striking and should be adequately explained and compared in the discussion.

Response:

Line 253-272 and 312-315. Thank you for the suggestion and we have made change in the text.

  1. Comment:

The size of Figure 1 should be increased for better visibility.

Response:

Line 201. We appreciate the suggestion and we have made changes in the text.

  1. Comment:

Line 158. The authors are combining ORs and confidence intervals from different criteria. This is not correct, and moreover, the upper limit of the confidence interval is incorrect. Please review this. Furthermore, the lower limit of this confidence interval is at the borderline; therefore, these results should be interpreted with caution. A similar situation occurs with criteria 5, 10, and 11(Table 2). This should be detailed in the abstract and explained and discussed further in the main text.

Response:

We agree with the reviewer's comment. We have written the sentence: “However, due to some lower limits of the confidence interval being at the borderline (criteria 5, 10, 11), these results should be interpreted with caution.

Discussion

  1. Comment:

Line 189, 211 and 218. Please consider the observations made about these results. Please consider the other aspects described above that need to be discussed. Additionally, more study limitations should be presented.

Response:

We appreciate the considerations and we have made improvements throughout the discussion, the limitations were better described in the line 305-315.

  1. Comment:

Line 244. This is not entirely accurate. Please review some studies on this topic published after the dissemination of the current classification.

Response:

Line 244-245. We appreciate the suggestion and we have made changes in the text.

Conclusions

  1. Comment:

Line 252. Please provide more specific details.

Response:

Line 328-329. We appreciate the suggestion and we have made changes in the text to improve understanding.

  1. Comment:

Some references are very outdated; therefore, it is recommended to replace them with current concepts.

Response:

We appreciate your suggestions and the references have been updated. 

Reviewer 3 Report

Comments and Suggestions for Authors

Title: Comparative study of periodontal parameters used in the diag-2 nosis of periodontitis in puerperae and its relationship with the 3 birth of preterm infants.

Manuscript ID: ijerph-2710284

The topic of the relationship between periodontitis and premature and/or low birth weight infants is a classic research topic in periodontal medicine. More than 300 international publications prove this. The authors delve into this topic by scientifically demonstrating a widely known fact. There are, however, errors and clarifications in the manuscript that are described below.

Introduction.

-Defining periodontitis according to a concept from 1998 (25 years) is not appropriate. Include aspects such as subgingival biofilm dysbiosis, bacteria as initiators of the inflammatory process but not as directly responsible for periodontal aggression..... must be highlighted in a current definition of periodontitis.

-The authors state: "Research and a systematic review of the possible relationship between periodontitis and prematurity have shown controversial results." In the last 5 years alone, there are 10 systematic reviews and/or meta-analyses on periodontitis (with or without treatment) and prematurity. The authors DO NOT include any of these studies. There are even highly cited narrative reviews on the subject, which the authors do not cite either:

Puertas A, Magan-Fernandez A, Blanc V, Revelles L, O'Valle F, Pozo E, León R, Mesa F. Association of periodontitis with preterm birth and low birth weight: a comprehensive review. J Matern Fetal Neonatal Med. 2018 Mar;31(5):597-602

The background provided by the authors is poor and inadequate.

-Again the authors state: "No standard diagnostic criteria for periodontitis are currently available due to the complex nature of the condition and its signs and symptoms." The authors cannot base their claims on publications that are nearly 20 years old. There are current clinical diagnostic criteria for periodontitis (they may be more or less fortunate), the latest classification of periodontal diseases and their conditions, published in 2018 jointly by the AAP and EFP.

-This statement:"The existing diagnostic criteria have not been considered in pregnant women, but only in patients in general", I don't know what it means. Clarify.

M&M

2.1

-Between what dates was the study carried out? The ethics committee has a date of 2011. Clarify.

-How many items of the STROBE checklist did this observational study fulfill? .This requirement should be included as additional material.

-The authors justify this sample size by: "...by the fact that the studied event (prematurity) is rare". This justification is not valid. The prevalence of premature and/or low birth weight is, depending on the country, between 9-11% of all births, this cannot be considered a rare pathology. Correct.

2.2

-Saying that the explorer was trained is NOT enough. Readers need to know, with intra and inter correlation data, the degree of correction in the exploration.

-The North Carolina periodontal probe is not PCPUNC 156, but 15. Correct.

-There are two very important variables that the authors have not collected. Tobacco consumption, closely related to periodontitis, and the height-weight of mothers, closely related to premature births. Failure to control these variables can cause bias in the analysis of the results. Although the BMI variable is described in the results section, it is here in M&M where all the study variables must be described correctly. The authors state that they collected "......and factors related to general health status". What factors related to general status? All confounding variables for prematurity must be known, described (correctly measured), for their control. Correct.

-What relationship does CAL have with reference 15? Clarify.

-What plate index did the authors use? Reference 16 of Ainamo and Bay is a bleeding index.

2.3

-What differences are there between criterion 8 and 11?. The presence of BOP is always necessary to be considered active periodontitis; this clinical criterion has always been necessary and required. Many of these criteria are not valid if they do not include this characteristic. For example, a PD = 5mm without BOP, may be a properly maintained periodontal patient.

-Criterion 8 corresponds to an index created by Lopez NJ et al. in 2002. It is fair to recognize its authorship and not that of another author (reference 7).

Lopez NJ, Smith PC, Gutierrez J. Periodontal therapy may reduce the risk of preterm low birth weight in women with periodontal disease: a randomized controlled trial. J Periodontol. 2002;73(8):911-24

-It is striking that the current one considered by the AAP and the EFP (2018 classification) has not been considered as a criterion. Clarify.

2.4

-Why did they use the non-parametric Mann-Whitney test, when they included a significant number of women in both groups? Parametric tests are more robust. Was checked if the periodontal variables did not follow a normal distribution? Clarify.

Results

-The results that the authors describe in PD and CAL raise serious doubts. Such marked differences in PD and the absence of differences in CAL, between both groups, makes me assume that it is gingivitis and not periodontitis, in the group of cases. Furthermore, this suspicion is confirmed by the greater gingival bleeding, almost significant in this group. These would be false periodontal pockets, because a true pocket of 4mm, with the gingival marginal close to the CEJ, would present a CAL of 3mm (always greater than 2mm).

-How can interpret that one of the laxest criteria, 13 (along with 14), has the lowest prevalence of periodontitis? Clarify.

Discussion

-First paragraph. This is correct: "All these criteria (5 to 8, 10, and 11) shared a PD parameter ≥ 4 mm, which signals a current inflammatory load" if they are accompanied by careful bleeding after probing. Not all of these indices are accompanied by positive BOP. Correct.

-The authors state in the fourth paragraph: "It should be noted that criteria 6, 7, 8, and 11 revealed a higher likelihood of periodontitis among mothers with preterm newborns, approximately two times higher when compared to those without periodontitis. This interpretation of the results, it is not correct. It would be that mothers with a diagnosis of periodontitis, according to these criteria, would have twice as likely to give birth to a premature baby.Clarify.

-I do not understand the justification given by the authors for not using the new classification of the American Academy of Periodontology. Precisely, it would have been a novel result of this study to use this criterion.

-It is missing that the authors try to explain how periodontitis can be a risk factor for prematurity.

Author Response

Introduction.

  1. Comment:

Defining periodontitis according to a concept from 1998 (25 years) is not appropriate. Include aspects such as subgingival biofilm dysbiosis, bacteria as initiators of the inflammatory process but not as directly responsible for periodontal aggression..... must be highlighted in a current definition of periodontitis.

Response:

Line 34-37. We appreciate the suggestion and we have made changes in the text.

  1. Comment:

The authors state: "Research and a systematic review of the possible relationship between periodontitis and prematurity have shown controversial results." In the last 5 years alone, there are 10 systematic reviews and/or meta-analyses on periodontitis (with or without treatment) and prematurity. The authors DO NOT include any of these studies. There are even highly cited narrative reviews on the subject, which the authors do not cite either: Puertas A, Magan-Fernandez A, Blanc V, Revelles L, O'Valle F, Pozo E, León R, Mesa F. Association of periodontitis with preterm birth and low birth weight: a comprehensive review. J Matern Fetal Neonatal Med. 2018 Mar;31(5):597-602

Response:

We appreciate the suggestion and we have made changes in the text. We included the studies below.

1.Tonetti MS, Greenwell H, Kornman KS. Staging and grading of periodontitis: Framework and proposal of a new classification and case definition. J Clin Periodontol. 2018; 45(Suppl 20): S149–S161.

  1. Puertas A, Magan-Fernandez A, Blanc V, Revelles L, O'Valle F, Pozo E, León R, Mesa F. Association of periodontitis with preterm birth and low birth weight: a comprehensive review. J Matern Fetal Neonatal Med. 2018 Mar;31(5):597-602.
  2. Karimi N, Samiee N, Moradi Y. The association between periodontal disease and risk of adverse maternal or neonatal outcomes: A systematic review and meta-analysis of analytical observational studies. Health Sci Rep. 2023 Oct 19;6(10):e1630. doi: 10.1002/hsr2.1630. PMID: 37867783; PMCID: PMC10587389.
  3. Vivares-Builes AM, Rangel-Rincón LJ, Botero JE, et al. Gaps in Knowledge About the Association Between Maternal Periodontitis and Adverse Obstetric Outcomes: An Umbrella Review. J Evid Based Dent Pract. 2018; 18(1):1-27. doi: 10.1016/j.jebdp.2017.07.006. Epub 2017 Jul 15.
  4. Oliveira LJ, Cademartori MG, Schuch HS, et al. Periodontal disease and preterm birth: Findings from the 2015 Pelotas birth cohort study. Oral diseases. 2021; 27(6), 1519-1527.
  5. Márquez-Corona MDL, Tellez-Girón-Valdez A, Pontigo-Loyola AP, et al. Preterm birth associated with periodontal and dental indicators: a pilot case-control study in a developing country. The Journal of Maternal-Fetal & Neonatal Medicine. 2021; 34(5), 690-695.

  1. Comment:

The background provided by the authors is poor and inadequate.

-Again the authors state: "No standard diagnostic criteria for periodontitis are currently available due to the complex nature of the condition and its signs and symptoms." The authors cannot base their claims on publications that are nearly 20 years old. There are current clinical diagnostic criteria for periodontitis (they may be more or less fortunate), the latest classification of periodontal diseases and their conditions, published in 2018 jointly by the AAP and EFP.

Response:

Line 57-58. Line 244-245. We appreciate the suggestion and we have made changes to the text. , they We included current studies using the new AAP classification.

  1. Comment:

-This statement:"The existing diagnostic criteria have not been considered in pregnant women, but only in patients in general", I don't know what it means. Clarify.

Response:

We appreciate the suggestion and we have made changes in the text. The diagnostic criteria were based on periodontal examinations performed on patients who were not pregnant.

M&M

2.1

  1. Comment:

-Between what dates was the study carried out? The ethics committee has a date of 2011. Clarify.

Response:

Line 70. We appreciate the suggestion and we have included in the text that the data for this study were collected between October 2011 and December 2012.

  1. Comment:

-How many items of the STROBE checklist did this observational study fulfill? This requirement should be included as additional material.

Response:

We appreciate the suggestion. The study contains the recommendations of STROBE.

  1. Comment:

-The authors justify this sample size by: "...by the fact that the studied event (prematurity) is rare". This justification is not valid. The prevalence of premature and/or low birth weight is, depending on the country, between 9-11% of all births, this cannot be considered a rare pathology. Correct.

Response:

We appreciate the suggestion and have removed the term rare. In Brazil, the 2023 rate was 11% According to the WHO, available on this website https://borntoosoonaction.org/

2.2

  1. Comment:

-Saying that the explorer was trained is NOT enough. Readers need to know, with intra and inter correlation data, the degree of correction in the exploration.

Response:

Line 110-112. We appreciate the comment, the calibration process was carried out in accordance with WHO recommendations, with 10 patients being examined at two different times, one week apart, at the Dental Clinic of the Federal University of Maranhão.

  1. Comment:

-The North Carolina periodontal probe is not PCPUNC 156, but 15. Correct.

 Response:

Line 121. We appreciate the suggestion and we have made changes in the text.

  1. Comment:

-There are two very important variables that the authors have not collected. Tobacco consumption, closely related to periodontitis, and the height-weight of mothers, closely related to premature births. Failure to control these variables can cause bias in the analysis of the results. Although the BMI variable is described in the results section, it is here in M&M where all the study variables must be described correctly. The authors state that they collected "......and factors related to general health status". What factors related to general status? All confounding variables for prematurity must be known, described (correctly measured), for their control. Correct.

Response:

Line 94-104 We appreciate the suggestion and we have made changes in the text.

  1. Comment:

What relationship does CAL have with reference 15? Clarify.

Response:

We appreciate the suggestion. We included a new reference. The reference is Lindhe J, Niklaus LP. Tratado de periodontia clínica e implantologia oral. 6. Rio de Janeiro: Guanabara Koogan, 2018, 1292 p.

  1. Comment:

-What plate index did the authors use? Reference 16 of Ainamo and Bay is a bleeding index.

Response:

We appreciate the comment.  We have maintained the reference because it addresses the gingival and the plaque index.

2.3

  1. Comment:

-What differences are there between criterion 8 and 11?. The presence of BOP is always necessary to be considered active periodontitis; this clinical criterion has always been necessary and required. Many of these criteria are not valid if they do not include this characteristic. For example, a PD = 5mm without BOP, may be a properly maintained periodontal patient.

Response:

Line 140-143 and Line 145-146. We appreciate the suggestion. The difference between criteria 8 and 11 is that criterion 8 also requires the presence of bleeding on probing in the same site. We agree that BOP presence is necessary to consider active periodontitis, however periodontal disease has been classified in different ways over the years, so we used different criteria from different times to try to compare them.

  1. Comment:

-Criterion 8 corresponds to an index created by Lopez NJ et al. in 2002. It is fair to recognize its authorship and not that of another author (reference 7).

Lopez NJ, Smith PC, Gutierrez J. Periodontal therapy may reduce the risk of preterm low birth weight in women with periodontal disease: a randomized controlled trial. J Periodontol. 2002;73(8):911-24

Response:

The Lopez criterion refers to criterion 11, as it does not mention bleeding on probing. Below is an excerpt from the article about the author's diagnostic criteria.

“The presence of 4 or more teeth with 1 or more sites with PD ≥ 4mm and with CAL ≥ 3mm at the same site was diagnosed as periodontal disease. “

  1. Comment:

-It is striking that the current one considered by the AAP and the EFP (2018 classification) has not been considered as a criterion. Clarify

Response:

We appreciate the reviewer's comment and as requested, we have included additional information about the 2018 Classification of Periodontal Diseases in the last paragraph (Tonetti et al., 2018). Furthermore, as this classification is very sensitive for our study sample since women have a low socioeconomic status and all have tooth loss, it was not possible to use it in this study as we would not have a comparison group, i.e. that is, women without periodontitis. Therefore, the following information was incorporated into the Discussion section:

As a final diagnostic criterion, it was proposed to use the classification of the American Academy of Periodontology and European Federation of Periodontology [1], however, according to this criterion, all women were diagnosed with periodontitis. This is due to the socioeconomic level of the sample and the number of missing teeth. Furthermore, no radiographic examination was performed because we could not expose the postpartum woman to radiation during breastfeeding and because of the difficulty for hospitalized women in traveling to a radiology center.

In addition to not using the current classification, another limitation that can be pointed out is the difficulty in discussing the clinical diagnostic parameters used, since no studies are found that compare the different diagnostic criteria for periodontal disease, except for the study by Gomes Filho. Among the strengths, it is essential to highlight that this is one of the few studies that made an adjustment of the analysis, an investigation of the specificity and sensitivity of the different criteria for diagnosing periodontitis, and the association of prematurity with the adoption of various definitions for periodontal disease, expanding the discussion of periodontal disease as an important exposure for prematurity.

2.4

  1. Comment:

-Why did they use the non-parametric Mann-Whitney test, when they included a significant number of women in both groups? Parametric tests are more robust. Was checked if the periodontal variables did not follow a normal distribution?

Response:

Line 153-155. We have rewritten the description of the statistical analysis to clarify this point. “The normality assumption was tested using the Shapiro-Wilk test. Most of the continuous variables violated the assumptions of normality, so the Mann-Whitney test was used to compare the periodontal parameters between groups.

Results

  1. Comment:

-The results that the authors describe in PD and CAL raise serious doubts. Such marked differences in PD and the absence of differences in CAL, between both groups, makes me assume that it is gingivitis and not periodontitis, in the group of cases. Furthermore, this suspicion is confirmed by the greater gingival bleeding, almost significant in this group. These would be false periodontal pockets, because a true pocket of 4mm, with the gingival marginal close to the CEJ, would present a CAL of 3mm (always greater than 2mm).

Response: We appreciate the reviewer's observation and understand the question. However, we inform that the findings of periodontal parameters refer to the means and respective standard deviations. Although the mean of each clinical parameter did not present a statistically significant difference between the case and control groups, the case group always presented the worst periodontal conditions, that is, a higher percentage of probing depth ≥ 4 mm, clinical attachment loss ≥ 3 mm, bleeding on probing and plaque index. The criteria that define the presence or absence of periodontitis combine clinical parameters, allowing greater specificity and avoiding false-positive diagnoses of the presence of periodontitis. To improve understanding of the text, this part of the Results section was rewritten, as follows:

Line 202. Figure 2 shows the mean and standard deviation of the periodontal variables. The case group presented the worst periodontal conditions compared to the control group, that is, a higher percentage of teeth with PD ≥ 4 mm, teeth with CAL ≥ 3 mm, BOP index, and plaque index. This difference was statistically significant for the percentage of teeth with PD ≥ 4 mm (p<0.006). Additionally, the multivariate regression analysis for CAL showed that arterial hypertension was related to higher levels of CAL in the evaluated sample (estimate = 3.49, SE = 1.02, p <0.001, Table 2).

  1. Comment:

-How can interpret that one of the laxest criteria, 13 (along with 14), has the lowest prevalence of periodontitis? Clarify.-

Response

We appreciate the suggestion. However, these criteria don't seem so laxest. Criterion 13 requires 50% of sites with BOP, that's a lot of sites. Classification 14 requires a CAL greater than 4mm, the average CAL found was less than 4. That's why we believe they didn't have such significant results.

Discussion

  1. Comment:

First paragraph. This is correct: "All these criteria (5 to 8, 10, and 11) shared a PD parameter ≥ 4 mm, which signals a current inflammatory load" if they are accompanied by careful bleeding after probing. Not all of these indices are accompanied by positive BOP. Correct.

Response:

Line 253-254 We appreciate the reviewer's comment. Only the criteria 5 to 8 are associated with the BOP. We change the text.

  1. Comment:

-The authors state in the fourth paragraph: "It should be noted that criteria 6, 7, 8, and 11 revealed a higher likelihood of periodontitis among mothers with preterm newborns, approximately two times higher when compared to those without periodontitis. This interpretation of the results, it is not correct. It would be that mothers with a diagnosis of periodontitis, according to these criteria, would have twice as likely to give birth to a premature baby. Clarify.

Response:

Line 252-254. We appreciate the reviewer's comment. It has already been rewritten.

  1. Comment:

-I do not understand the justification given by the authors for not using the new classification of the American Academy of Periodontology. Precisely, it would have been a novel result of this study to use this criterion.

Response:

We appreciate the reviewer's comment and as requested, we have included additional information about the 2018 Classification of Periodontal Diseases in the last paragraph of discussion (line 305 to 311) (Tonetti et al., 2018). Furthermore, as this classification is very sensitive for our study sample since women have a low socioeconomic status and all have tooth loss, it was not possible to use it in this study as we would not have a comparison group, i.e. that is, women without periodontitis. Therefore, the following information was incorporated into the Discussion section:

As a final diagnostic criterion, it was proposed to use the classification of the American Academy of Periodontology and European Federation of Periodontology [1], however, according to this criterion, all women were diagnosed with periodontitis. This is due to the socioeconomic level of the sample and the number of missing teeth. Furthermore, no radiographic examination was performed because we could not expose the postpartum woman to radiation during breastfeeding and because of the difficulty for hospitalized women in traveling to a radiology center.

In addition to not using the current classification, another limitation that can be pointed out is the difficulty in discussing the clinical diagnostic parameters used, since no studies are found that compare the different diagnostic criteria for periodontal disease, except for the study by Gomes Filho. Among the strengths, it is essential to highlight that this is one of the few studies that made an adjustment of the analysis, an investigation of the specificity and sensitivity of the different criteria for diagnosing periodontitis, and the association of prematurity with the adoption of various definitions for periodontal disease, expanding the discussion of periodontal disease as an important exposure for prematurity.

  1. Comment:

-It is missing that the authors try to explain how periodontitis can be a risk factor for prematurity.

Response:

Line 247-252. Line 34-37. We appreciate the suggestion and we have made changes in the text.

Round 2

Reviewer 2 Report

Comments and Suggestions for Authors

The authors have significantly improved the quality of the manuscript; however, two issues still need to be addressed.

1. The conclusions in the abstract and the main text should indicate that they are based on the study's limitations. The term "strong" should be eliminated considering the logistic regression results, and caution is recommended in describing the conclusions.

2. On the other hand, the authors present the calibration process but do not describe the number of clinicians who participated, and the results of such calibration measured by any statistical test.

These adjustments to the manuscript will enhance the quality of the study.

Comments on the Quality of English Language

moderate editing

Author Response

Dear reviewers,

We would like to thank you again for your attention and suggestion to our study. We are sure that all the notes enriched the manuscript and made it more fluid to read. We, the authors, analyze the suggestions one by one and highlight them in blue in the text.

Comments and Suggestions for Authors

The authors have significantly improved the quality of the manuscript; however, two issues still need to be addressed.

  1. The conclusions in the abstract and the main text should indicate that they are based on the study's limitations. The term "strong" should be eliminated considering the logistic regression results, and caution is recommended in describing the conclusions.

Response:

Thank you very much. We accepted the reviewer's suggestion and made changes to the abstract and conclusions.

  1. On the other hand, the authors present the calibration process but do not describe the number of clinicians who participated, and the results of such calibration measured by any statistical test.

Response:

We appreciate the reviewer's observation and inform that although there was training for the clinical periodontal examination with an experienced periodontal specialist, only one examiner, also a specialist in periodontics, performed the clinical examination throughout the entire research, making the examination even more reliable, with no need to estimate the Kappa index. Therefore, we added the following information in the Material and Method section, as follows:

"A single investigator previously trained, periodontics specialist, and blinded to the condition of birth performed the periodontal examination throughout the research. The calibration process was carried out in accordance with WHO recommendations, with 10 patients being examined at two different times, one week apart, at the Dental Clinic of the Federal University of Maranhão [13].".

These adjustments to the manuscript will enhance the quality of the study.

Comments on the Quality of English Language                   moderate editing

  1. Moderate editing of English language required
    Response:

Thank you very much. We accepted the reviewer's suggestion and a rigorous review of the English language was carried out in the text.

Reviewer 3 Report

Comments and Suggestions for Authors

Most of the 22 improvement proposals have been adequately answered, included, and corrected by the authors. However, proposition 8 must be completed to increase the internal validity of the study. It refers to the fact that the authors must include in the methodology, inter- and intra-explorer agreement values as they are metric variables.

Author Response

Dear reviewers,

We would like to thank you again for your attention and suggestion to our study. We are sure that all the notes enriched the manuscript and made it more fluid to read. We, the authors, analyze the suggestions one by one and highlight them in blue in the text.

Comments and Suggestions for Authors

Most of the 22 improvement proposals have been adequately answered, included, and corrected by the authors. However, proposition 8 must be completed to increase the internal validity of the study. It refers to the fact that the authors must include in the methodology, inter- and intra-explorer agreement values as they are metric variables.

Response:

We appreciate the reviewer's observation and inform that although there was training for the clinical periodontal examination with an experienced periodontal specialist, only one examiner, also a specialist in periodontics, performed the clinical examination throughout the entire research, making the examination even more reliable, with no need to estimate the Kappa index. Therefore, we added the following information in the Material and Method section, as follows:

"A single investigator previously trained, periodontics specialist, and blinded to the condition of birth performed the periodontal examination throughout the research. The calibration process was carried out in accordance with WHO recommendations, with 10 patients being examined at two different times, one week apart, at the Dental Clinic of the Federal University of Maranhão [13].".